# Severe Multiorgan Failure Following Yellow Fever Vaccination

**DOI:** 10.3390/vaccines8020249

**Published:** 2020-05-26

**Authors:** Cristina Domingo, Judith Lamerz, Daniel Cadar, Marija Stojkovic, Philip Eisermann, Uta Merle, Andreas Nitsche, Paul Schnitzler

**Affiliations:** 1Robert Koch Institute, Centre for Biological Threats and Special Pathogens, 13353 Berlin, Germany; Domingo-CarrascoC@rki.de (C.D.); NitscheA@rki.de (A.N.); 2Department of Internal Medicine IV, University Hospital of Heidelberg, 69120 Heidelberg, Germany; Judith.Lamerz@med.uni-heidelberg.de (J.L.); Uta.Merle@med.uni-heidelberg.de (U.M.); 3Bernhard Nocht Institute for Tropical Medicine, National Reference Centre for Tropical Pathogens, 20359 Hamburg, Germany; danielcadar@gmail.com (D.C.); philip.eisermann@gmail.com (P.E.); 4Center for Infectious Diseases, Tropical Medicine, University Hospital of Heidelberg, 69120 Heidelberg, Germany; Marija.Stojkovic@med.uni-heidelberg.de; 5Center for Infectious Diseases, Virology, University Hospital of Heidelberg, 69120 Heidelberg, Germany

**Keywords:** yellow fever virus, vaccination, viscerotropic disease, multiorgan failure

## Abstract

Background: The yellow fever (YF) vaccination is recommended by the WHO for people traveling or living in endemic areas at risk for yellow fever infections in Africa and South America. Although the live attenuated yellow fever vaccine is a safe and efficient vaccine, rare serious adverse events after vaccination have been reported. Case presentation: We present the case of a 74-year-old male with multiorgan failure after yellow fever vaccination for a trip to Brazil. The patient required admission to the intensive care unit with a prolonged stay due to severe organ dysfunction. Five days after the YF vaccination, the patient experienced nausea, vomiting, diarrhea, and general illness. Three days later he sought medical attention and was transferred to the University Hospital Heidelberg with beginning multiorgan failure and severe septic shock, including hypotonia, tachypnea, thrombopenia, and acute renal failure the same day. Within one week after vaccination, antibodies against YF virus were already detectable and progressively increased over the next two weeks. Viral RNA was detected in serum on the day of admission, with a viral load of 1.0 × 10^5^ copies/mL. The YF virus (YFV) RNA was also present in tracheal secretions for several weeks and could be detected in urine samples up to 20 weeks after vaccination, with a peak viral load of 1.3 × 10^6^ copies/mL. After 20 weeks in the ICU with nine weeks of mechanical ventilation, the patient was transferred to another hospital for further recovery. Conclusions: The risk for severe adverse events due to the YF vaccination should be balanced against the risk of acquiring a severe YF infection, especially in elderly travelers.

## 1. Background

Yellow fever virus (YFV) is a member of the genus *Flavivirus*, family Flaviviridae, mainly transmitted by *Aedes* sp. mosquitoes. Despite the availability of a safe and effective vaccine, YF remains a public health issue in 34 African countries and 13 South American countries. Although some YFV-infected people have an asymptomatic course of infection, most patients show symptoms like fever, headache, nausea, muscle pain, backache, vomiting, jaundice, and bleeding from the mouth, nose, eyes, or stomach. The disease might progress into full hemorrhagic syndrome with multiorgan failure. Treatment of YF is only supportive. Jaundice accompanied by increased liver enzymes is a leading symptom for a severe cause of the disease and is significantly associated with risk of death [1]. Renal failure is also a clinical manifestation of a severe and fatal YF outcome.

The prevention against YF could be achieved by administering the live attenuated 17D vaccine, providing lifelong immunity against the disease in most vaccinated individuals. Although an excellent safety profile of the vaccine exists, some severe adverse events following YF-immunization (YFV-AEFI) occurred. YF-AEFI includes severe allergic reactions (e.g., anaphylaxis), neurological disease termed YF vaccine associated neurotropic disease (YEL-AND), and a serious, frequently fatal, multisystemic illness: YF vaccine associated viscerotropic disease (YEL-AVD). YEL-AVD has been estimated at a frequency of about 0.3–0.4 per 100,000 doses distributed in vaccinees [2], and it has been described in the setting of primary vaccination in people without pre-existing YFV immunity. Elderly males (≥56 years), young women (19–34 years), and individuals with thymus disorders have been identified as risk groups for the development of YEL-AVD [3,4]. The clinical presentation of YEL-AVD includes high-grade fever (≥38 °C for more than 24 h) and other signs such as nausea, vomiting, malaise, myalgia, arthralgia, diarrhea, and dyspnea in the early phase, resembling acute natural YF infection. YEL-AVD patients frequently present jaundice, thrombocytopenia, elevation of hepatic enzymes, total bilirubin, and creatinine. As the disease progresses patients may show cardiovascular instability, hemorrhage, and in some cases respiratory and/or renal failure, resembling wild type YF. YEL-AVD has a very short incubation period (2–7 days after vaccination), and a fatality rate of over 50%. The identification and characterization of suspected cases of adverse events after YFV vaccination is important to assess the safety of the vaccine. Here we describe a confirmed case of YEL-AVD in a 74-year-old traveler.

## 2. Case Presentation

In December 2017, a 74-year-old male with arterial hypertension and a cured prostate carcinoma planning his travel to Brazil was vaccinated against YF (Stamaril, Sanofi Pasteur, Val de Reuil, France). He had no history during a previous stay in YF endemic areas. Five days post- vaccination the patient experienced nausea, vomiting, diarrhea, and general malaise. Two days later he presented to a hospital with the beginnings of multiorgan failure and was immediately transferred to the University Hospital Heidelberg. On admission at the ICU, the patient showed signs of septic shock, including disseminated intravascular coagulation, hepatitis, acute renal failure, and cardiorespiratory insufficiency. At the initial presentation, the patient showed signs of severe septic shock, including hypotonia, lactate acidosis (70 mg/dL), tachypnea, thrombopenia (35/nL), septic encephalopathy, acute renal failure (serum creatinine 4.77 mg/dL, GFR according to MDRD 11.8 mL/min/1.73 qm), and elevated inflammation markers CRP and PCT (Table 1). According to the criteria defined by the Brighton Collaboration Viscerotropic Working Group, six of seven major criteria for the definition of viscerotropic disease applied, including hepatic, renal, musculoskeletal, respiratory, platelet disorder, and hypotension, confirming level 1 of diagnostic certainty for viscerotropic disease [5]. Septic shock was treated according to international standards. Respiratory insufficiency was observed with progressive systemic hypoxemia and clinical exhaustion, which resulted after a short period of high-flow oxygen therapy in intubation and mechanical ventilation. Due to severe acute respiratory distress syndrome and the inability of reaching lung-protective ventilation (default setting of 6 mL/kg PBW, Horowitz index < 100 mmHg), ventilation was started in a prone position, and nitric oxide ventilation was administered for four days. CT imaging showed no remarkable results.

Specific YF laboratory analyses demonstrated the presence of IgM and IgG antibodies (IgM/IgG YFV IIF, EUROIMMUN, Lübeck, Germany) as well as neutralizing antibodies [5], with a PRNT 90 titer of 1:13 in a serum sample obtained on day 8 after vaccination. A neutralization antibody titer of 1:10 or higher is considered a surrogate of protection [6]. In the same sample, a high viral load of YFVs RNA was observed with 1.0 × 10^5^ copies/mL (Figure 1). A diagnosis was made of suspected YEL-AVD. Other concomitant infections or causes for the symptoms were not identified, i.e., hepatitis A, B, C, E; HIV; herpes simplex virus; varicella zoster virus; Legionella, Leptospira; and Aspergillus. Due to recurrent inflammation, including respiratory-associated pneumonia, several antibiotics were applied. We saw signs of rhabdomyolysis (CK 68,000 U/L) with clinically indurated right gluteus maximus without need of surgical intervention. Severe paralytic ileus was treated conservatively. Over the course, cytomegalovirus (CMV) retinitis resulted in transient blindness; an MRT imaging showed in spite of atrophy of the optical nerve no further central impairment. Due to renal failure, an intermittent hemodialysis was necessary. The viral load in serum dropped to 3.0 × 10^2^ copies/mL over the next weeks, and YFV neutralizing antibody titers increased over time, reaching peak titers at 50 days after vaccination of ≥1:320. Serial samples of serum, plasma, tracheal secretion, and urine were analyzed for YFV by PCR. In serum, viral RNA was detectable at days 5–17 after vaccination, in plasma at day 36, in tracheal secretion at day 22, and between days 50–88 after vaccination. In urine, an initial viral load of 1.3 × 10^6^ copies/mL was detected in a sample one week after vaccination and, moreover, remained positive more than 100 days after vaccination (Figure 1), definitively confirming the causality of the YF vaccine [4]. Deep sequencing had been employed in order to obtain the full genome of the YFV strain detected in the patient. Complete genome sequencing of the patient-derived isolate, designated as BNI-YFV-17 (GenBank MG922934), did not reveal changes when compared to the sequence of the YFV 17D vaccine (GenBank AVT50843.1).

Altogether the patient was treated for 20 weeks on ICU, ventilated for nine weeks, received hemodialysis for 10 weeks, and was transferred to a another hospital for recovery. At discharge, the multiorgan failure was completely resolved, and only mild anemia was still present. Persistent health problems included impairment in mobility and autonomy in daily life, dysphagia, i.e., the need for recannulation of the tracheostoma to prevent aspiration and a reduced visus with shades of gray and silhouettes.

## 3. Discussion and Conclusions

We successfully treated a 74-year-old man who experienced multiorgan failure eight days after YF vaccination. Clinical examination and laboratory tests excluded other potential conditions, e.g., bacterial and viral infections. The clinical features, their temporal association with the YF vaccination, and the identification of the YF vaccine virus by RT-PCR and sequencing, suggest a causal relation between the illness and YF vaccination. Multiorgan failure after the YF vaccination is a rare condition, however the multisystemic clinical picture developed by the patient is similar to previous described cases of YEL-AVD after vaccination [7,8]. The advanced age of the patient appears to have been a risk factor for developing YEL-AVD after a primary YF vaccination [3,4].

The YF viral load detected shortly after vaccination reflects the severity of the disease and can be used as a prognostic marker for the progression of the disease in the patient [9]. Low-level transient viremia is detectable in vaccinees 4–6 days after vaccination. However, higher levels have been reported only in wild-type infections and severe YF vaccination adverse events [10]. The high level of viral load detected 10 days after vaccination in serum, and the detection of YFV viral RNA 17 days after vaccination in serum and 36 days after vaccination in plasma, could be considered as an unequivocal criterion for definite YEL-AVD. It has been observed that the average viral load in the urine of vaccinees with adverse events is 8.8 × 10^2^ genome equivalents/mL and ranges between 30 to 10^4^ genome equivalents/mL, without any difference regarding the day of sample collection [11]. In our patient, 10^5^ genome equivalents/mL more than one week after vaccination were observed. Moreover, YF RNA was detectable in the urine of the patient for 20 weeks after vaccination; much longer than in the serum as previously demonstrated in cases of suspected YF-AEFI [11]. The finding could be related to YFV replication, which may occur in the kidney and be correlated with the renal failure in our patient who required dialysis. Remarkably, YFV RNA was detected in respiratory samples (bronchial lavage, tracheal secretion) until nine weeks after vaccination and consistent with the symptomatology of the patient who presented with multiorgan failure, severe ARDS, circulatory failure, hepatitis, intravascular coagulation, and severe paralytic ileus. Over the course of hospitalization, a multiresistant *Pseudomonas aeruginosa* was detected in tracheal secretions.

Our patient showed signs of reduced vision, noticed 5–6 weeks after vaccination. A YF vaccination-related, Vogt–Koyanagi–Harada-like disease has been recently reported [12]. Although there are some similarities between both cases, in our patient it was unclear when it first occurred, since the patient was long-term mechanically ventilated, had a tracheostoma which limited verbal communication, and suffered from delirium. The clinical examination had no remarkable findings, except a light anisocoria, which might be due to opioid therapy. A cranial MRT only showed an atrophy of the left opticus nerve. Viral load for CMV was positive and CMV-induced retinitis assumed, thus the patient was treated with ganciclovir. There was no significant improvement of vision after discharge.

A single dose of YF vaccine provides long-lasting protection in the majority of vaccinees [2], and the YF vaccine’s safety profile in routine practice is favorable [13]. However, vaccination against YF has been associated with rare cases of viscerotropic and neurotropic disease [14,15,16,17]. The vaccines currently in use contain heterogeneous virus subpopulations, which could explain some differences affecting the safety of the vaccines. The presence of viral genomic mutations and the selection of specific viral subpopulations have been reported in cases of YF-AEFI [18]. Genomic sequencing of the patient-derived isolate did not reveal any changes with respect to the YFV 17D strain.

Khromava et al. [4] updated the estimates of YFV-adverse visceral disease and neurotropic disease events; the reporting rates of serious adverse events were significantly higher among vaccinees aged ≥60 years than among those 19–29 years of age. A review concluded that the evidence for risk for the elderly remains limited, and vaccination should be based on a risk–benefit analysis [19]. In a study by Lindsey et al. [2], on average, 3.8 severe adverse events were reported per 100,000 vaccine doses, severe adverse events increased with increasing age with a rate of 6.5 per 100,000 in persons aged 60–69 years and 10.3 for persons ≥ 70 years. A longer viremia after YF vaccination has been observed in older vaccinees [20], and the acute immune response to YF vaccination both humoral and CD8+ T cell response is affected in this group, which could result in a reduced capacity to clear the vaccinal virus. The higher risk of adverse events after YF vaccination observed for people over 60 years of age may be rooted in immune senescence during aging—a process that may have contributed to the specific case in this report. It has been proposed that an inactivated YFV vaccine would be of benefit for this population [21].

This case highlights the importance of continued education for physicians providing traveler advice regarding the risks and benefits of YF vaccination, particularly for older travelers to avoid YEL-AVDs. Severe vaccine-derived adverse effects have to be discussed prior to the YF vaccination. Indication for a YF vaccination should consider the specific geographical area endemic for YF and not only the country of destination. The risks of acquiring YF in bigger cities are considerably lower compared to remote areas of the countries.

## Figures and Tables

**Figure 1 vaccines-08-00249-f001:**
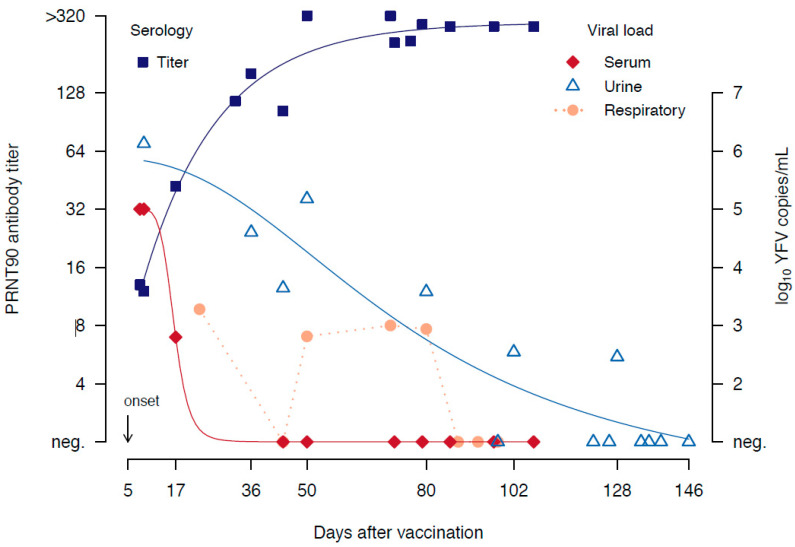
Timeline of yellow fever neutralizing antibody titers and yellow fever viral load observed in serum, urine, and respiratory samples (bronchoalvelar lavage, tracheal secretion). Regression lines indicate a general trend.

**Table 1 vaccines-08-00249-t001:** Results of clinical chemistry parameters, values out of normal range are marked in bold.

Parameter	Normal Range	Days after Yellow Fever Vaccination					
		Day 8	Day 13	Day 15	Day 22	Day 40	Day 134
creatinine (mg/dL)	0.6–1.4 mg/dL	4.77	3.7	3.2	2.45	6.03	1.09
GFR (MDRD, mL/min/1.73 qm)	>60	11.8	-	-	-	-	-
GOT (U/L)	<46 U/L	607	2056	1747	255	75	45
GPT (U/L)	<50 U/L	473	-	286	87		45
gGT (U/L)	<60 U/L	131	-	152	241	152	46
AP (U/L)	40–130 U/L	119	-	130	235	-	115
bilirubin (mg/dL)	<1.0 mg/dL	1.1	-	6.3	4.3	-	-
LDH (U/L)	<342 U/L	853	-	-	-	454	-
CRP (mg/L)	<5 mg/L	143.5	146	149	67.5	39.8	3.7
PCT (ng/mL)	<0.05	41.08	55.76	28.11	-	-	-
leucocytes (/nL)	4–10/nL	11.5	11.9	10.8	14	14	9.4
quick (%)	70–125%	61.6	68.2	69	81	95.5	112
INR	<1.2	1.25	1.18	1.18	1.09	1.03	0.96
thrombocytes (/nL)	150–440/nL	35	13	33	153	461	358
lactate (mg/dL)	<16 mg/dL	69.6	-	-	-	-	-
CK (U/L)	<190 U/L		68,329	33,753	1261	268	-
pH	7.37–7.45	7.42	-	-	-	-	-
pO2 (mmHg)	-	76	-	-	-	-	-
pCO2 (mmHg)	35–45 mmHg	27	-	-	-	-	-

## Data Availability

All data generated or analyzed during this study are included in this article.

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
