# Peer review of "Severe Multiorgan Failure Following Yellow Fever Vaccination"

_vaccines, 2020, doi:10.3390/vaccines8020249_

Round 1
Reviewer 1 Report
Thank you for inviting me to review the case report "Severe multiorgan failure associated with yellow fever vaccination". The case describes a vaccination for yellow fever where the attenuated viral vaccination led to infection and multi-organ failure. The authors present a well written and well described case presentation and discuss the need to weigh the relative risks of vaccination, travel and infection, particularly in the context of people of advanced aged. While I have no major concerns with the manuscript, could the authors discuss or comment on the need/availability/feasibility of a non-attenuated alternative vaccinations.
Author Response
We fully agree with the reviewer to include a comment on the need and feasibility of a non-attenuated alternative vaccination. See revised manuscript, lines 174 – 175: It has been proposed that an inactivated YFV vaccine would be of benefit for this population [26].
See also reference #26:
Jonker EF, Visser LG, Roukens AH. Advances and controversies in yellow fever vaccination. Ther Adv Vaccines. 2013; 1: 144-52.
Reviewer 2 Report
Please see the attached pdf file: Review Paul Schnitzler_1.pdf

Author Response
The reviewer is absolutely right, the title has been changed according to the reviewer’s suggestion: Severe multiorgan failure following yellow fever vaccination
2.
We thank the reviewer for the alignment of the sequences. After careful analysis of our sequencing data, we concluded, that there is no mutation in the patient-derived isolate, see lines 102 – 106: “Deep sequencing has been employed in order to obtain the full genome of the YFV strain detected in the patient. Complete genome sequencing of the patient-derived isolate designated as BNI-YFV-AND/17 (GenBank MG922934), did not reveal changes when compared to the sequence of the YFV 17D vaccine (GenBank AVT50843.1).” See also lines 159 – 162: “The presence of viral genomic mutations and the selection of specific viral subpopulations have been reported in cases of YF-AEFI [22]. Genomic sequencing of the patient-derived isolate did not reveal any changes with respect to the YFV 17D strain.”
3.
See comment on mutation above
4.
The reviewer is right regarding infections with other flaviviruses. In the case reported, no need for a differential diagnosis regarding other flavivirus was needed. The case occurred in Germany where other flaviviruses that could resemble the same clinical presentation (i.e. dengue) are not present and moreover, the patient did not have a recent travel history to other flavivirus endemic areas. The causal relationship with the vaccine is demonstrated by the long presence of high titers of the vaccine YF strain (and no other flavivirus) in the body of the patient. The serological diagnosis of flaviviral infections has as drawback the presence of cross reactions among the members flaviviruses, and is only reserved for those cases in which a molecular diagnosis is not possible
5.
We fully agree with the reviewer to provide more information about the neutralisation assay. This information is now provided in the revised manuscript, see reference #6, Reinhardt et al. for the PRNT90 method:
Reinhardt B, Jaspert R, Niedrig M, Kostner C, L’age-Stehr J. Development of viremia and humoral and cellular parameters of immune activation after vaccination with yellow fever virus strain 17D: a model of human flavivirus infection. J Med Virol 1998; 56:159–67.
The sentence with the titer has been changed according to the suggestion of the reviewer, see lines 95 – 97: “The viral load in serum dropped to 3.0 x 102 copies/ml over the next weeks, and YFV neutralizing antibody titers increased over time reaching peak titers at 50 days after vaccination of ≥ 1:320.”
6.
At the time of submission the sequence of the patient-derived isolate to GenBank, there was not a clear case definition for the patient. It has been decided to submit a provisional title which will be updated once the manuscript is released online.
Minor points
All minor points have been corrected according to the comments of the reviewer. We thank the reviewer for pointing out to define some missing abbrevations, these are included in the revised manuscript, see lines 182 – 189:
Abbreviations: YFV: Yellow fever virus; YEL-AVD: yellow fever vaccine associated viscerotropic disease; YF-AEFI: yellow fever adverse event following immunization; ICU: intensive care unit; GOT: glutamate oxaloacetate transaminase; GFR: glomerular filtration rate; MDRD: modification of diet in renal disease; CRP: C-reactive protein; PCT: procalcitonin; CT: computer tomography; MRT: magnetic resonance tomography; LDH: lactate dehydrogenase; AP: alkaline phosphatase; gGT: gamma glutamyl transferase; GPT: glutamate pyruvate transaminase; INR: international normalized ratio; CK: creatinine kinase; ARDS: acute respiratory distress syndrome; RNA: ribonucleic acid

Reviewer 3 Report
1. Specific YF laboratory analyses demonstrated the presence of IgM and IgG antibodies as well as neutralizing antibodies above the protection limit (titer 1:13) in a serum sample obtained on day 8 after vaccination. In the same sample, a high viral load of YFVs RNA was observed with 1.0 x 105 copies/ml (Fig. 1).
Due to the few commercial assays for the detection of IgM and IgG anti YFV it would be appreciable to indicate the tests used and better explain the seroneutralizazion protocol.
2. Serial samples of serum, plasma, tracheal secretion and urine were analyzed for YFV by PCR. In serum, viral RNA was detectable during weeks 1 to 3 after vaccination, 6 weeks after vaccination in plasma, in tracheal secretion in weeks 4 to 10 after vaccination. In urine, an initial viral load of 1.3 x 106 copies/ml was detected in a sample one week after vaccination and remained positive up to 20 weeks after vaccination (Fig. 1)
Given the important viral load in patient biological samples It would be important to perform the analysis of quasispecies in different samples and not only in the sera sample as performed.
Therefore, it is necessary to define the presence of the same mutation or others through NGS of different biological matrix.
3. However, higher levels have been reported only in wild-type infections and severe YF vaccination adverse events. The high level of viral load detected 10 days after vaccination in serum, and the detection of YFV viral RNA 17 days after vaccination in serum and 36 days after vaccination in plasma could be considered as an unequivocal criterion for definite YEL-AVD. It has been observed that the average viral load in vaccinees with adverse events is 8.8 x 102 genome equivalents/ml and ranges between 30 to 104 genome equivalents/ml, without any difference regarding the day of sample collection [10].
Further references to these statements are needed. In fact these points are the basis of the rationale for work.
Author Response
We fully agree with the reviewer to indicate the serological test that was used and to explain the neutralization protocol. See revised manuscript, lines 83 – 86: “Specific YF laboratory analyses demonstrated the presence of IgM and IgG antibodies (IgM/IgG YFV IIF, EUROIMMUN, Lübeck, Germany) as well as neutralizing antibodies [6], with a titer of 1 : 13. This titer is above the protective level of ≥ 1 : 10, which was determined in a serum sample obtained on day 8 after vaccination.”
The information for the neutralisation assay is now provided in the revised manuscript, see reference #6, Reinhardt et al. for the PRNT90 method:
Reinhardt B, Jaspert R, Niedrig M, Kostner C, L’age-Stehr J. Development of viremia and humoral and cellular parameters of immune activation after vaccination with yellow fever virus strain 17D: a model of human flavivirus infection. J Med Virol 1998; 56:159–67.
2.
After careful analysis of our sequencing data, we concluded, that there is no mutation in the patient-derived isolate, see lines 102 – 106: “Deep sequencing has been employed in order to obtain the full genome of the YFV strain detected in the patient. Complete genome sequencing of the patient-derived isolate designated as BNI-YFV-AND/17 (GenBank MG922934), did not reveal changes when compared to the sequence of the YFV 17D vaccine (GenBank AVT50843.1).” See also lines 159 – 162: “The presence of viral genomic mutations and the selection of specific viral subpopulations have been reported in cases of YF-AEFI [22]. Genomic sequencing of the patient-derived isolate did not reveal any changes with respect to the YFV 17D strain.”
3.
We thank the reviewer for the comment concerning viral loads and more information regarding this issue has been included into the revised manuscript, see lines 131 – 136: “The high level of viral load detected 10 days after vaccination in serum, and the detection of YFV viral RNA 17 days after vaccination in serum and 36 days after vaccination in plasma could be considered as an unequivocal criterion for definite YEL-AVD [14]. It has been observed that the average viral load in in urine of vaccinees with adverse events is 8.8 x 102 genome equivalents/ml and ranges between 30 to 104 genome equivalents/ml, without any difference regarding the day of sample collection [15].
See also newly added reference #14:
Gershman MD, Staples JE, Bentsi-Enchill AD, et al. Viscerotropic disease: case definition and guidelines for collection, analysis, and presentation of immunization safety data. Vaccine. 2012; 30: 5038–5058.

Round 2
Reviewer 2 Report
Reviewer: The manuscript from Christina Domingo et al. was resubmitted and the modified version shows some improvements.
The authors should consider the following points to finalize their manuscript:
Authors response to reviewer:
- The reviewer is right regarding infections with other flaviviruses. In the case reported, no need for a differential diagnosis regarding other flavivirus was needed. The case occurred in Germany where other flaviviruses that could resemble the same clinical presentation (i.e. dengue) are not present and moreover, the patient did not have a recent travel history to other flavivirus endemic areas. The causal relationship with the vaccine is demonstrated by the long presence of high titers of the vaccine YF strain (and no other flavivirus) in the body of the patient. The serological diagnosis of flaviviral infections has as drawback the presence of cross reactions among the members flaviviruses, and is only reserved for those cases in which a molecular diagnosis is not possible
Reviewer: I am still irritated why the authors have not excluded presence of antibodies to other flaviviruses (i.e. USUV, WNV) as a possible underlying reason for the vaccine failure described in the patient. Can the authors exclude that there is no antibody enhancing effect by cross reacting ab? The causal relationship was not criticized; the authors should look for reasons of vaccine failure especially when the three centers are claiming that flaviviruses are emerging in Germany.
Reviewer: Fig 1: trend lines are incorrect especially for serum titers. Additionally, authors should explain the negative respiratory diagnostic observed on ~ day 44.
Reviewer: Lanes 83-85: is it: ,with a PRNT90 titer of 1:13,
What do the authors mean with: protective level of ≥ 1:10… which was determined on day 8.
In the same sample they identified the highest viral load in serum and urine. How can they claim that the NT titer is protective? The protective level is with reference to what kind of test PRNT90, PRNT50 ?
Please specify and rephrase.
Reviewer: No information on the vaccine administered is given in the case report. Please specify.
Author Response
Reviewer #2
As the reviewer acknowledges, the clinical picture of the patient did not resemble an Usutu infection (West Nile was still not present in Germany at that time) and a possible infection was unlikely during the winter time (December in the case of our patient). We consider that the detection of antibodies against Usutu or West Nile would no add a better understanding of the case, since the presence of cross-reactivities in the serological assays among flaviviruses would only add a confounding factor in this case.
Moreover, even when it is true that the immunological crosstalk between heterologous flaviviruses may increase the risk of severe disease through a mechanism of antibody-dependent enhancement of infection, this has not been demonstrated for yellow fever vaccination after millions of doses applied in areas highly endemic for other related flavivirus (i.e. dengue). Adverse events after vaccination have not been associated to the presence of previous immunity against other flaviviruses, while other risk factors have been clearly identified (age, immunosuppression or presence of thymus disease). In our case, we strongly consider that the age of the patient posed a risk to experience an adverse event after vaccination to him.
2.
We have carefully considered the remark concerning Figure 1, but the rationale for omitting trend lies is uncertain. In addition, we think that trend lines in this plot are a useful visual aid that enhances readability and allow the different measured parameters to be compared on a single graph.
The negative result on day 44 for the respiratory sample is most probably depending on the sample itself (quality, collection).
3.
>Lanes 83-85: “with a titer of 1 : 13. This titer is above the protective level of ≥ 1 : 10, which was determined in a serum sample obtained on day 8 after vaccination.
We have rephrased the sentence for clarity as suggested by the reviewer. Now it reads:
“with a PRNT 90 titer of 1 : 13 in a serum sample obtained on day 8 after vaccination. A neutralisation antibody titer of 1:10 or higher is considered a surrogate of protection.” See bold sentence in revised manuscript. In addition we have added reference #7 in the revised manuscript (Monath TP, Cetron MS, McCarthy K, et al. Yellow fever 17D vaccine safety and immunogenicity in the elderly. Hum Vaccin. 2005; 1: 207-214).
The presence of high viral loads of the vaccine virus on day 8 after vaccination in serum and urine demonstrates that the replication of the virus exceeded the clearance capacity of the patient at this time point. Innate immunity may play a role in in controlling the virus at this early time point, which is impaired in elderly persons due to senescence of the immune system putting these persons at a higher risk of adverse events. When a person gets in contact with the virus by a mosquito bite and has a pre-existing immunity to the virus (titer more or equal to 1:10), the situation might be different, as the viral load is very small and a robust immune response can be mounted in short time.
4.
We fully agree with the reviewer to provide information on the vaccine administered. See revised manuscript, lines 63 – 65: “In December 2017 a 74-year old male with arterial hypertension and a cured prostate carcinoma planning his travel to Brazil was vaccinated against YF (Stamaril®, Sanofi Pasteur, Val de Reuil, France).”
Reviewer 3 Report
Although it is a single clinical case report, it is well described and It could be considered a useful clinical and laboratory support to consider and evaluate the effects of vaccination against YFV.
Author Response

(The authors gave the same response as above.)
